# A Machine Learning Evaluation of the Effects of South Africa's COVID-19 Lockdown Measures on Population Mobility

**Albert Whata** [1,*] and **Charles Chimedza** [2]

1. School of Natural and Applied Sciences, Sol Plaatje University, Kimberley 8301, South Africa
2. School of Statistics and Actuarial Science, University of the Witwatersrand, Johannesburg 2050, South Africa; charles.chimedza@wits.ac.za
* Correspondence: albert.whata@spu.ac.za

**Abstract:** Following the declaration by the World Health Organisation (WHO) on 11 March 2020, that the global COVID-19 outbreak had become a pandemic, South Africa implemented a full lockdown from 27 March 2020 for 21 days. The full lockdown was implemented after the publication of the National Disaster Regulations (NDR) gazette on 18 March 2020. The regulations included lockdowns, public health measures, movement restrictions, social distancing measures, and social and economic measures. We developed a hybrid model that consists of a long-short term memory auto-encoder (LSTMAE) and the kernel quantile estimator (KQE) algorithm to detect change-points. Thereafter, we utilised the Bayesian structural times series models (BSTSMs) to estimate the causal effect of the lockdown measures. The LSTMAE and KQE, successfully detected the changepoint that resulted from the full lockdown that was imposed on 27 March 2020. Additionally, we quantified the causal effect of the full lockdown measure on population mobility in residential places, workplaces, transit stations, parks, grocery and pharmacy, and retail and recreation. In relative terms, population mobility at grocery and pharmacy places decreased significantly by $-17{,}137.04\%$ ($p$-value = 0.001 < 0.05). In relative terms, population mobility at transit stations, retail and recreation, workplaces, parks, and residential places decreased significantly by $-998.59\%$ ($p$-value = 0.001 < 0.05), $-1277.36\%$ ($p$-value = 0.001 < 0.05), $-2175.86\%$ ($p$-value = 0.001 < 0.05), $-370.00\%$ ($p$-value = 0.001< 0.05), and $-22.73\%$ ($p$-value = 0.001 < 0.05), respectively. Therefore, the full lockdown Level 5 imposed on March 27, 2020 had a causal effect on population mobility in these categories of places.

**Keywords:** causal effect; encoder–decoder; kernel quantile estimator; long-short term memory; population mobility; reconstruction error

## 1. Introduction

On March 11, 2020, the World Health Organisation (WHO) declared that the global COVID-19 outbreak had become a pandemic [1]. Consequently, a national state of disaster was declared by the government of South Africa on of 15 March 2020 [2]. When the outbreak worsened, the government ordered all South Africans into a full lockdown. The full lockdown was effective for 21 days from 26 March 2020. The full lockdown was implemented after the publication of the National Disaster Regulations (NDR) gazette on 18 March 2020 [3]. The regulations or measures contained in the gazette were applicable for the duration of the full lockdown. These drastic regulations or measures that were imposed on the public included lockdowns, public health measures, movement restrictions, social distancing, and social and economic. For example, the nation-wide lockdown which was initially set for 21 days ending 16 April 2020, required that everyone except those providing essential services stayed at home. People were only allowed to leave their homes for urgent food shopping and medical treatments. The lockdown measure was imposed to fundamentally disrupt the chain of transmission of the corona virus and to stop the spread of the virus thereby saving South African lives. Although the lockdown was viewed as the

best response from a public health perspective, the economic impact was devastating for ordinary South African households and businesses [4].

In this paper, we seek to identify change-points [5] using the Google COVID-19 Community Mobility Reports [6] and the South African government COVID-19 measures contained in the Government Measures Dataset provided by ACAPS [7]. Additionally, we study how these government interventions affect population mobility in areas including workplaces, residential, transit stations, parks, grocery and pharmacy, and retail and recreation. The aim is to quantify the causal effects of the South African government interventions on population movements in these areas. The literature on change-points informed part of our approach in evaluating the causal effects of the government measures. For example, [8] indicates that for historical data, control charts have traditionally been used to detect changes. Moreover, for large datasets, change-point detection is more desirable than control charts as the approach can well-characterise smaller changes [9]. The authors mention that single change-point methods have applied classical statistical thresholding algorithms based on the mean and variance on offline data as well as non-parametric tests for changes in distributions. In addition, other statistical methods estimate the probability of a change-point occurring by utilizing Bayesian priors that incorporate time-dependent information.

We developed a hybrid model that consists of a long-short term memory auto-encoder (LSTMAE) and the kernel quantile estimator (KQE) algorithm to detect change-points. We used the LSTMAE algorithm because it has been shown to perform better in anomaly or intrusion detection [10–14]. There are some advantages to using LSTMs. For example, LSTMs can be used in sequences of varying lengths [15] without making any assumptions on the number of previous points that are needed for making predictions. LSTMs are structured to exploit temporal dependencies in sequential data, and they do not assume any functional form between the outcome variables and regressors or explanatory variables [16].

LSTMs are a variant of recurrent neural networks (RNNs). RNNs are a very powerful tool in deep learning [17]. According to [18], RNNs outperform conventional machine learning methods that include $k$-nearest neighbors (KNN) and support vector machines (SVM) because they contain "memory" that captures past information regarding what has been calculated and they can also learn long-range patterns. On the other hand, we use the KQE to determine a threshold that is used for detecting anomalies in the reconstruction errors obtained from the LSTMAE. The KQE is desirable because we do not want to assume a parametric form for the distribution of the reconstruction errors [19]. This means that the KQE offers flexibility over parametric estimators as we can choose from several classes of functions where we assume the reconstruction errors to belong. In addition, the KQE expresses the univariate distribution of reconstruction errors as a finite mixture and thus, gives a smooth distribution from which to estimate quantiles [20].

After detecting the change-point(s) we then create a Bayesian structural time series model (BSTSM), to predict a counterfactual and then measure the causal effect of the South African government interventions such as a lockdown (change-point) on population mobility. The BSTSM is implemented in the *R* package, CausalImpact [21] utilising the COVID-19 Community Mobility Reports by Google [6]. A study of the causal effect of the interventions will provide insights on the effect of the government's measures and hopefully assist those making critical decisions to combat COVID-19 or any other possible future pandemic.

Change-point detection entails finding the location in a sequence of observations where the statistical properties change [22]. Detecting change-points is important in many different application areas. Several supervised and unsupervised techniques that can be utilised for change-point detection in time series data were surveyed by [23]. Change-point detection has primarily been used to model and predict time series in several application areas such as climatology [24], bioinformatic applications [25], finance [26], medical imaging [27], speech [28] and image analysis [29].

Change-point analysis can be employed to evaluate the effect of an intervention using synthetic control methods (SCM) for comparative case studies [30]. According to [31], the use of SCM for comparative case studies involves comparing units that are subjected to an event or intervention of interest to one or more units that are not exposed. This means that comparative case studies are only possible when some units are exposed to an intervention whilst others are not exposed. Thus, change-points can be used to separate units that are exposed to an intervention and units that are not exposed. When investigating the effect of a policy or intervention, identifying change-points in each dataset on interventions is very important. This is because the change-point analysis should verify that a change-point has indeed occurred at the time or point of intervention. Thus, at a change-point, we can estimate the average treatment effect of a change-point or intervention. The challenge in change-point analysis is to come up with an algorithm to automatically detect changes in the properties of sequences for us to make the appropriate decisions. This is because change-points in a sequence can be described as "rare events", like anomalies which make it harder for the classification problem to detect it, as the dataset will be heavily imbalanced. The points are changes only at that temporal context and not as independent points. Hence, this problem is difficult to solve using general classification algorithms. Because LSTMs have memory within their structure, they are better suited to capture the patterns inside the sequences. The behavioural change in the sequence at any temporal context will also have patterns among them. Thus, LSTMs make a reasonable option to solve the change-point detection problem as they have the capability to learn the patterns in sequences. While associated means or variances can be obtained, we specifically focus on detecting the positions and number of the change-points. After detecting and verifying the existence of change-points, a model is fit that can predict the counterfactual utilising the pre-intervention time series and then compare the predicted (counterfactual) to the actual times that are recorded after the intervention. The BSTSMs that can be implemented in CausalImpact R package [21] are then used to estimate the average treatment effect of an intervention (change-point).

## 2. Review of Literature

In this section, a review of literature and the related work is presented.

### 2.1. Causal Inference the Counterfactual Approach

Causal inference has been studied in Statistics [32,33], Econometrics [34] and Biostatistics [35]. The studies have focused mostly on a setup where there is a binary treatment, and the Rubin Causal Model (RCM) or the potential outcomes framework is often used [36]. The basic element of causal inference is that each unit in a large population is characterised by the potential outcomes $Y_i(0)$ and $Y_i(1)$ [37]. Besides, the difference, $Y_i(1) - Y_i(0)$, gives the unit-level, treatment effect. If we let $Z \in \{0,1\}$ be a binary indicator for treatment, with $Z_i = 0$ if unit *i* received the control and $Z_i = 1$ if unit *i* received the active treatment, then $Y_i(1)$ is the outcome if unit *i* receives the active treatment and $Y_i(0)$ is the outcome if unit *i* receives the control. It should be noted that $Y_i(1)$ and $Y_i(0)$ can never be observed at the same time on the same unit, [36] refers to this as the Fundamental Problem of Causal Inference. Because of the fact that we can never realise $Y_i(0)$ and $Y_i(1)$ at the same time on the same unit, [36] states that the average causal effect then becomes the typical measure of a causal effect. Calculation of the average causal effect involves exposing some units in the population to treatment 1, giving information about $E(Y_i(1))$ and some units to treatment 0, giving information about $E(Y_i(0))$. Since both outcomes are observable, the average treatment effect is estimated by: $\tau = E(Y_i(1)) - E(Y_i(0))$.

There are basically two important assumptions linked to causality that need to be considered and these are the endogeneity [38] and the Ignorable Treatment Assignment assumptions [39]. Endogeneity occurs when the error term *(e)* is correlated a regressor *(x)* [38]. A variety of conditions can lead to violations of this assumption, but one important case occurs when key variables are excluded from the model. This means, there exist some

other variables not included in the model that are correlated with both the dependent and the independent variable(s). Ref. [40] also point out that not taking omitted variables into account will create biased treatment effects in observational/quasi-experimental designs, thereby, affecting the estimation of accurate causal effects. Thus, it is important to be able to articulate all the reasons why a participant is assigned to a particular treatment. Failure to identify all the reasons will result in an endogeneity problem.

The counterfactual approach is important in causal inference and analysis, it imagines that individuals may occupy multiple causal states, and each has multiple potential outcomes, one for each causal state [41]. The counterfactual framework relies on the assumption of randomisation conditional on the covariates ("unconfoundedness") stated below [42]:

$$(Y_i(0), Y_i(1)) \perp Z_i | X_i \qquad (1)$$

The assumption states that the pair of counterfactual outcomes, $(Y_i(0), Y_i(1))$, is independent of $Z_i$ (treatment variable) given the covariates $X_i$ [40]. The assumption is also known as the Ignorable Treatment Assignment Assumption [39]. Consequently, causal models usually encompass this assumption.

### 2.2. Related Work

The causal effects of the COVID-19 restrictions imposed by different federal states in Germany were investigated by [43]. Their causal analysis was aimed at contributing to the broader effort of scientists to understand how the Covid-19 pandemic was spreading as well as probe the causal role of political interventions such as social distancing. The Difference-in-Differences method was used to identify causal effects of COVID-19 policies by [44]. They proposed a difference-in-differences (DD) design for estimating causal effects in the COVID-19 context because governments implement certain policies differently. A DD design compares outcomes before and after a given COVID-19 related policy, to how the outcomes changes in an area that did not implement the policy. Care must be taken when carrying out DD analysis because of the common trends assumption violations, which may appear in COVID-19 contexts [44]. The DD designs may be affected by time lags between exposure to SARS-CoV-2 and the recorded infections, as well as variations in person-to-person transmission, and the possibility that the effects of the policies maybe different over time.

The maximum likelihood approach that is implemented in the change-point R package [22], was used to detect change points in population mobility trends in India [45]. The change-points were caused by the Covid-19 interventions imposed by the government of India. We extend the work by [45] and propose a hybrid model that utilises an LSTM auto-encoder and a kernel quantile estimator [17,19] to automatically detect change-points. Besides, we used the BSTSM to estimate the causal effect of a change-point. According to [21], the following assumptions must be met when using the BSTSM to estimate the treatment effects: (i) there exists a time series that is not affected by the intervention, i.e., the control (ii) the relationship between time series affected by the intervention, i.e., the response and the control is stable during the post-intervention period. When these assumptions are met, the BSTSM is used to construct a time-series model, perform posterior inferences on the counterfactual, and then return a treatment effect for a given response and control time series. Parametric models such as the difference-in-difference may have restrictive assumptions, which may make them harder to implement [46]. Therefore, we make the contributions listed below:

1. We develop a hybrid model that consists of a long-short term memory auto-encoder (LSTMAE) and the kernel quantile estimator (KQE) algorithm to automatically detect change points from a time series or a sequence of values,
2. We compare the change points detected by our proposed model, the long-short term memory auto-encoder (LSTMAE) that is combined with a kernel quantile estimator

   (LSTME and KQE) to the maximum likelihood algorithm, Bayesian Analysis models
   and linear regression models
3.   We estimate the causal effect of a change-point or intervention using the Bayesian
   structural time series model (BSTSM) that has fewer assumptions.

The rest of this paper is organized as follows: Section 3 describes briefly the materials and methods used, Section 4 describes the experiments, Section 5 presents the results and analysis, and Section 6 provides a discussion of the results and concludes the paper.

## 3. Materials and Methods

### 3.1. Data

The publicly available datasets used in this study are the ACAPS COVID-19 Government Measures Dataset [7] and the Google COVID-19 Community Mobility Reports [6]. Comprehensive information reported by countries, that details the different governments' interventions to control the spread of COVID-19 is captured in the ACAPS COVID-19 Government Measures Dataset. The dataset reports on the following categories of interventions; lockdowns, movement restrictions, social distancing, social and economic measures, and public health measures. In addition, Government measures of different severity are included. The Google COVID-19 Community Mobility Reports show graphs of population mobility trends over time, across six different categories of places such as groceries and pharmacies, transit stations, retail and recreation, workplaces, parks, and residential. These reports compare the changes in the number of visits and duration of stay to baseline days. The baseline days are based on normal movement values for a particular weekday expressed as the median. This median value is taken over five weeks (3 January 2020 to 6 February 2020) (https://ourworldindata.org/covid-mobility-trends) (accessed on 22 November 2021). Thus, population mobility measures the number of visits and duration of stay for a particular day relative to a baseline value for that day of the week from 3 January 2020, to 6 February 2020. This means that measuring changes for a particular day and comparing them to a normal weekday is important since people's routines during weekdays differ from their routines during weekends. However, we do not consider changes in population mobility that are a result of seasonal variations because population mobility to grocery and pharmacy as well as retail and recreation places usually increases during month-ends and paydays. In this paper, we are only interested in changes that can be explained by the pandemic and the government's interventions, and not changes that reflect seasonal movements. Consequently, we utilise South Africa's government interventions and community mobility reports as a case study and detect all the change-points between 15 February 2020 and 31 July 2020. Thereafter, we restrict ourselves to change-points that correspond to national lockdowns. We then evaluate the effects of these national lockdowns on population mobility across all the six categories of places.

### 3.2. Long Short-Term Memory Networks

Long short-term memory (LSTM) networks which were introduced by [47] can learn long-term dependencies through recurrently connected subnets known as memory blocks. LSTMs are a special kind of RNN that can detect change-points. A recurrent neural network attempts to model a time or sequence-dependent variable [48]. The most important function of an LSTM is to forget irrelevant parts of the previous state, selectively update a current state and then output certain parts of the present state that are relevant to future states. This solves the vanishing gradient problem [47] common in RNNs by updating a state then propagating forward some parts of the state that are relevant to future states. Therefore, LSTMs become far more efficient than RNNs as there is no extended chain of back-propagation as seen in RNNs. More details on the LSTM can be found in [47–49]. The idea of using LSTMs for detecting change-points is to build an LSTM encoder–decoder structure, and then apply it to sequential data to reconstruct the input data.

### 3.3. Change-Point Detection

Given a time series, $x_{1:n} = \{x_1, \ldots, x_n\}$, a single change-point is said to occur when the properties of $\{x_1, \ldots, x_\tau\}$ and $\{x_{\tau+1}, \ldots, x_n\}$ at a time or a point, $\tau \in \{1, \ldots, N-1\}$ are statistically different in the mean, variance or regression structure [50]. Besides, the single change-point ideas can be used for detecting multiple change-points. Therefore, if there are several change-points, $m$, and corresponding locations, $\tau_{1:m} = (\tau_1, \ldots, \tau_m)$, then location of each change-point is an integer between 1 and $n-1$. If $\tau_0 = 0$ and $\tau_{m+1} = n$, then for ordered change-points: $\tau_i < \tau_j \Leftrightarrow i < j$, $m+1$ segments are created by splitting the $m$ change-points. The $i$-th segment will contain the data $y_{(\tau_{i-1}+1):\tau_i}$. In addition, a set of parameters summarises each segment. For example, the set of parameters $\{\theta_i, \phi_i\}$ is associated with the $i$-th segment, where $\phi_i$ are nuisance parameters that should be evaluated when estimating the parameters, $\theta_i$ that contain the changes. Thereafter, we determine the number of segments that represent the data and evaluate the location as well as the number of change-points that are related to each segment.

### 3.4. Detecting Change-Points Using LSTMAE and KQE

Figure 1 shows a schematic representation of the proposed hybrid LSTM auto-encoder (LSTMAE) neural network model for detecting change-points. In Figure 1, we show an LSTMAE that is organised into an architecture called an Encoder–Decoder [14] LSTM. The LSTMAE can support input sequences of variable length and predict or output sequences of variable length. The motivation for using the LSTM autoencoder model in detecting change-points is that we want to use the weights obtained from the "normal" sequence to represent the training data well. The same weights are then used on the test data and the prediction errors are then used to identify change-points.

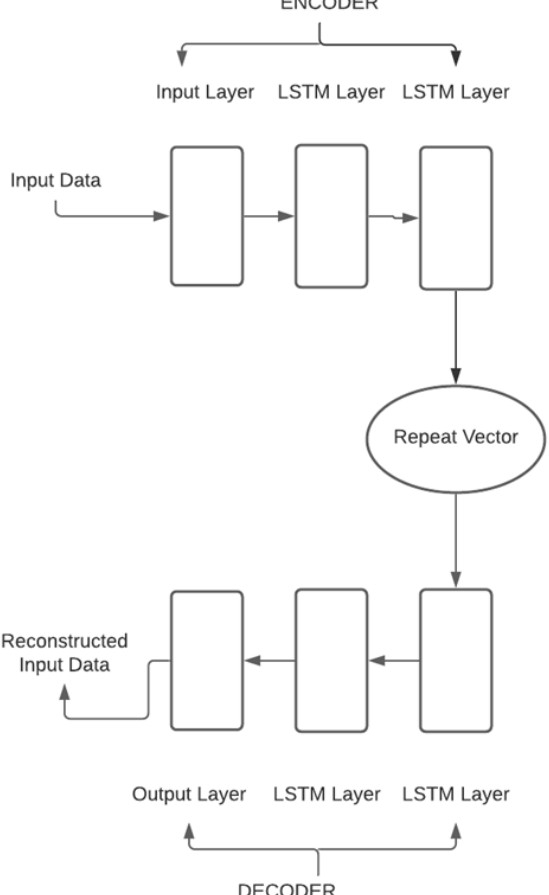

**Figure 1.** Schematic representation of the long-short term memory (LSTMAE) network.

We let $\{x_1, \ldots, x_n\}$, be a time series or sequence of data. We refer to this sequence as the normal sequence. Thereafter, we train the autoencoder on the normal data so that we reconstruct a new sample. The input is compressed by the encoder into a latent representation and then the decoder reconstructs the input from the latent space representation. We compare the input and the reconstructed output to calculate the prediction or reconstruction errors, $e_i = ||\hat{x}_i - x_i||$, and obtain a vector of errors $\{e_1, \ldots, e_n\}$, for $i \in (1, \ldots, n)$. Our change-point detection technique uses these reconstruction errors. The distribution of these errors is assumed to be a multivariate Gaussian distribution [17]. The authors argue that the assumption may be difficult to satisfy in practice because the distribution of the reconstruction errors is often not known. In this paper a nonparametric method that uses the kernel quantile estimator [17,19] is proposed to estimate the threshold $\tau$ that is then used to detect an anomaly using the reconstruction errors. To estimate $\tau$ we consider the ordered error or reconstruction vectors $\{e_1 \leq e_2 \leq \cdots \leq e_n\}$ from a sample of size $m$. Given a density function, $K$, that is symmetric about zero, then the threshold $\tau$ can be described as a kernel quantile estimator that can be evaluated using (3) [19]:

$$\tau_p = \sum_{i=1}^{n} \left[ \int_{\frac{i-1}{n}}^{\frac{i}{n}} \frac{1}{h} K\left( \frac{t-p}{h} \right) dt \right] e_{(i)} \tag{2}$$

where the bandwidth, $h > 0$, is an important parameter that regulates the extent of the smoothing employed to a sample of size $m$, and $p$: $0 < p < 1$. Ref. [17] point out that there are several versions for the approximation of $\tau_p$ and the choice of $K$ has little effect on the estimation performance. The kernel function that will be used in this paper is the widely used Gaussian of zero mean and unit variance [51] and it is given by:

$$K(u) = \frac{1}{\sqrt{2\pi}} exp\left( \frac{-u^2}{2} \right). \tag{3}$$

The assumption about the distribution of the reconstruction errors is not required when using a quantile kernel estimator [51]. Moreover, the smoothness of the density estimate is significantly influenced by the bandwidth, in contrast to $K$ that does not have any influence on the smoothness of the density estimate. Good results can be obtained by using different functions as $K$ is not very sensitive to the shape of the estimator [51]. However, choosing an efficient methodology for computing $h$, for an observed data sample is very important in practice [51]. This is because the bandwidth significantly affects the shape of the corresponding estimator. For example, an under-smoothed estimator will result from using a small bandwidth $h$ and an over-smoothed estimator further away from the function to be estimated will result from using a large bandwidth $h$. Ref. [17,19] apply an asymptotically optimised bandwidth $h_{opt}$, which is expressed as:

$$h_{opt} = \left( \frac{p(1-p)}{m+1} \right)^{\frac{1}{2}} \tag{4}$$

We use the value of $\tau_p$ to detect a change-point in a sequence or time series. At another time-step $t$, a reconstruction error is calculated by using $||\hat{x}_t - x_t||$, for a new observation, $x_t$, reconstruction $\hat{x}_t$ of $x_t$, that is predicted from the trained model. A candidate change-point, $x_t$ is identified if $e_t > \tau_p$. After detecting a candidate change-point and its position, the causal effect of the change-point or intervention on population mobility is then evaluated.

There are other methods that are used to detect change points such as; 1. the maximum likelihood approach [22]. We implement the likelihood ratio approach for detecting change-points by using the change-point R package [22], 2. Bayesian analysis [52], which is implemented in R using the bcp package [53], and 3. the structural changes in linear regression models [54], that are implemented using the algorithm, breakpoints() that is available in the R package strucchange [53].

### 3.5. Estimating Causal Effects Using Bayesian Structural Time-Series Models

After detecting the change-points, we use the BSTSMs to assess the effect of an intervention [21]. In this case, the difference between actual times series and the counterfactual time series estimates after the intervention, is used to assess the causal effect of an intervention. This difference gives the semi-parametric Bayesian posterior distribution for the causal effect. The counterfactual is estimated at the point where there is a change-point, because at that point the time series before and after the change-point differ in the mean or variance. The BSTSM framework uses available prior knowledge about the model parameters in determining the counterfactual. Additionally, the framework also uses state-space time series models which include linear regressions of the contemporaneous predictors. The Bayesian approach calculates the posterior distribution of the counterfactual time series given the pre-intervention time series [21].

Structural time-series models, for example, the BSTSMs are state-space models which can be represented by the following equations [21]:

$$y_t = Z_t^T \alpha_t + e_t, \tag{5}$$

$$\alpha_{t+1} = T_t \alpha_t + R_t \eta_t, \tag{6}$$

where $e_t \sim N(0, \sigma_t^2)$ and $\eta_t \sim N(0, Q_t)$ are independent of all other unknowns. The observation equation that links the observed data $y_t$ to a latent $d$-dimensional state vector $\alpha_t$ is shown in (10). The state equation that regulates the progression of the state vector $\alpha_t$ over time is shown in (11). Following [21], we consider, $y_t$ as a scalar quantity, $Z_t$ as a $d$-dimensional output vector, $T_t$ as a $d \, x \, d$ transition matrix, $R_t$ as a $d \, x \, q$ control matrix, $e_t$ as a scalar observation error with noise variance $\alpha_t$, and $\eta_t$ as a $q$-dimensional system error with a $q \, x \, q$ state-diffusion matrix $Q_t$, where $q \leq d$. The BSTSM framework is used to learn these parameters and thereafter, the Markov chain Monte Carlo (MCMC) technique, and a Gibbs sampler are employed to perform posterior inference. An estimate of the causal effect is calculated as the difference between the counterfactual (predicted) and the actual response during the post-intervention period.

The BSTSMs use the state-space models defined in (10) and (11) as well as flexible Bayesian priors to fit a time series model pre-intervention [21]. The counterfactual is then predicted using the fit model. The estimation of the causal effect and statistical tests of significance of an intervention using the BSTSM can be done in R using the CausalImpact R package. The vignette from (https://cran.r-project.org/web/packages/CausalImpact/vignettes/CausalImpact.html) (accessed on 25 November, 2020). together with the paper by [21] give detailed discussions of BSTSMs. We perform Bayesian structural time-series causal inference using the experiments described below:

### 4. Experiments

We used the ACAPS dataset [7] and the Google COVID-19 Community Mobility Reports [6] dataset. We deployed our proposed model (LSTMAE) described in Section 3.5 to identify change-points in all the six categories of places. The concept for using LSTMAE for change-point detection was taken from successful applications of the LSTME in anomaly detection [11,55–57]. Python 3 was used to create and train our LSTMAE neural network model. TensorFlow [58] was used as our back end and Keras [59] was utilised as our core model development library. Subsequently, datasets for training and testing our LSTMAE were defined. The data were split into a first part which is the "normal" data or training data, without any change-point. This dataset spans from 15 February 2020 to 26 March 2020. These dates include the point where the government of South African declared a national state of disaster on 15 March 2020. The test data was from 27 March 2020, to 30 April 2020, which included the point where the government ordered all South Africans into a national lockdown for 21 days. For each of the 6 categories of places, we plotted the whole data set

from 15 February 2020, to 26 March 2020, to visually check for the existence of any possible change-points.

We normalised and reshaped the data into a suitable input format for LSTMAE neural network. LSTMs cells expect a three-dimensional tensor as input. The LSTMAE neural network architecture used is shown in Figure 1. The first set of layers called the encoder creates the compressed representation of the input data. Thereafter, the compressed representational vector is distributed across the decoder's time steps by a repeat vector layer. The reconstructed input data is produced by the decoder's final output layer. The efficient Adam optimiser [60] is used for training the model. The mean absolute error (MAE) is utilised as the loss function. The model is trained for 500 epochs.

Using the kernel quantile estimator (3), we determine a threshold value $\tau_p$ for identifying change-points. The reconstruction errors are calculated in the training dataset as well as in the test dataset to determine where the error values exceed the threshold $\tau_p$ and thus, detect a change-point $m$. Once a value of $m$ has been positively identified as a candidate change-point which represents a government intervention, the CausalImpact R package to evaluate the average causal effect of a government intervention on our outcome variable. The CausalImpact R is also used to test for the statistical significance of the average causal effect.

We compare the change-points detected by using reconstruction errors from the LSTM autoencoder and the kernel quantile estimator to the change-points detected by the change-point R packages which detect changes in the mean or variance or both. This is done to determine whether our proposed deep learning approach detects the same change-points as detected by other methods available in practice.

## 5. Results and Analysis

### 5.1. Change-point Detection Using Different Algorithms

Figure 2 shows the time series mobility trends for residential areas, transit stations, parks, workplaces, grocery and pharmacy, and retail and recreation.

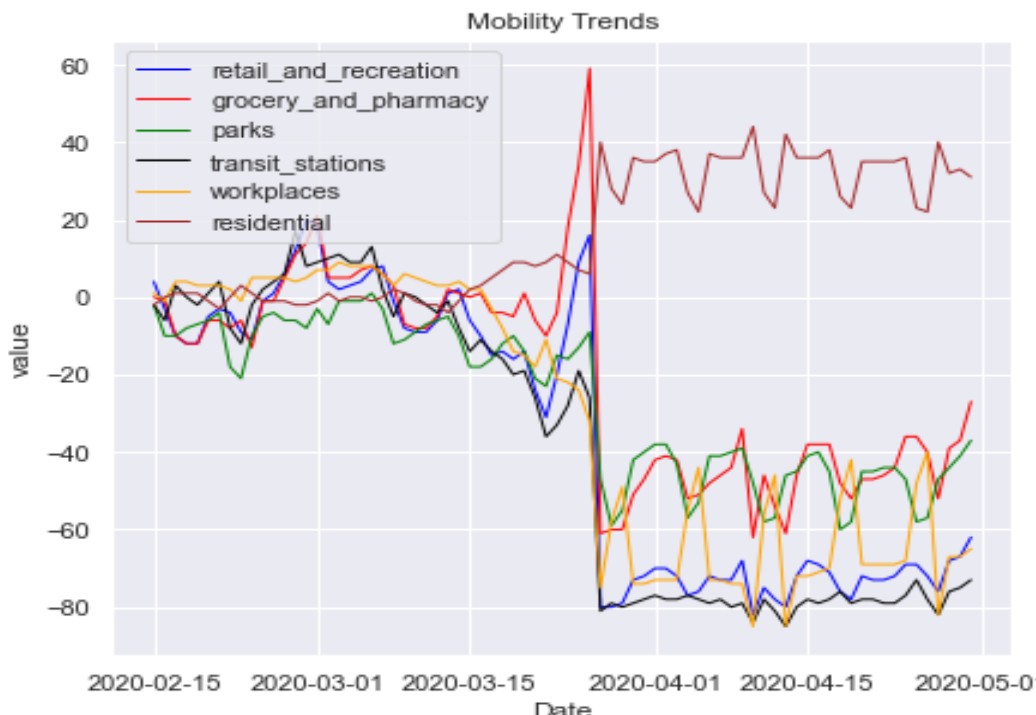

**Figure 2.** Plots of the datasets for the 6 categories of places.

The plots show that the mobility trends before and after the first intervention, a full lockdown imposed by the government of South Africa on 27 March 2020, are indeed different indicating that 27 March 2020, is a candidate change-point.

The LSTMAE and KQE algorithm was trained for each of the 6 categories of places in order to detect the change-points and their positions, using the community mobility dataset described in Section 3.1. The dataset is based on the COVID-19 Community Mobility Reports by Google [6] dataset and its constitutes the changes in the number of visits and duration of stay corresponding to all the days from 15 February 2020, to 30 April 2020, inclusive. The LSTMAE and KQE algorithm was trained using the procedure outlined in Section 4. Figures 3–8 below shows the change-points that were detected using the LSTMAE and KQE algorithm, for each of the six categories of places.

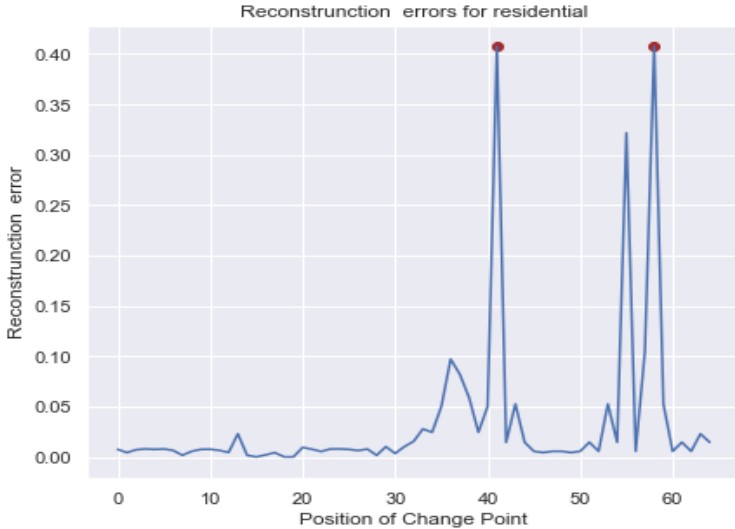

**Figure 3.** Reconstruction errors for residential.

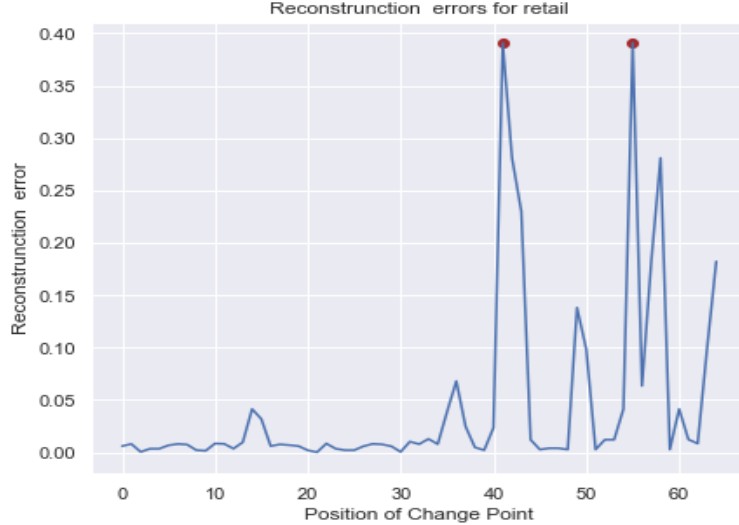

**Figure 4.** Reconstruction errors for grocery and pharmacy.

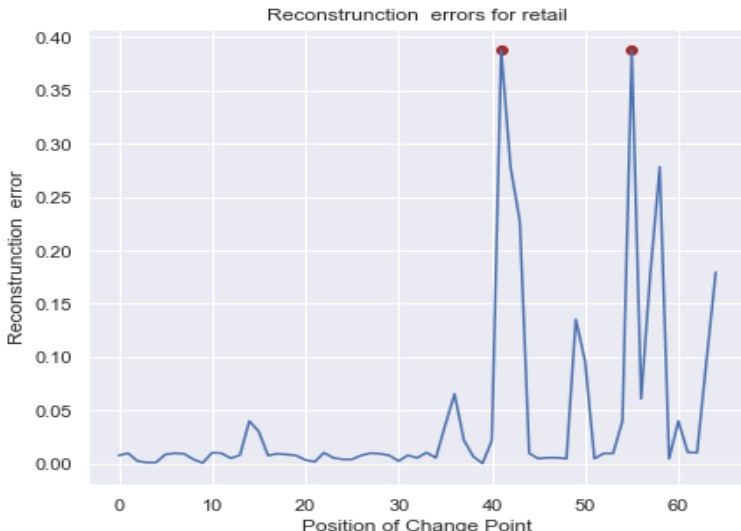

**Figure 5.** Reconstruction errors for retail and recreation.

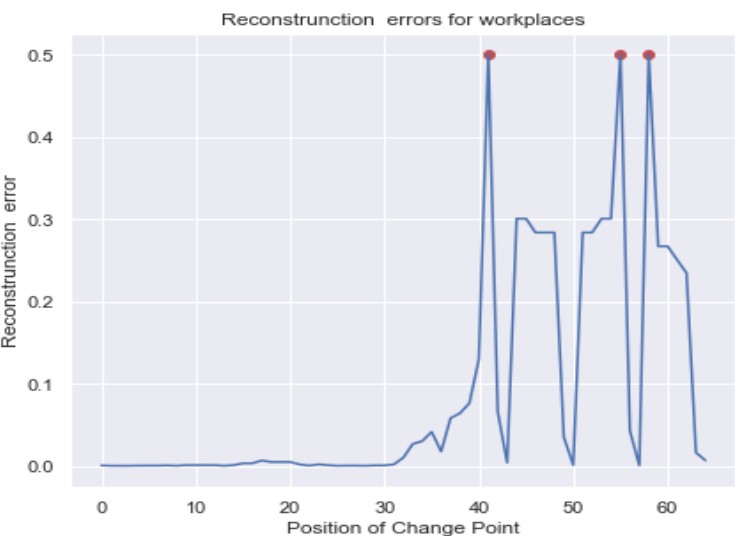

**Figure 6.** Reconstruction errors for workplaces.

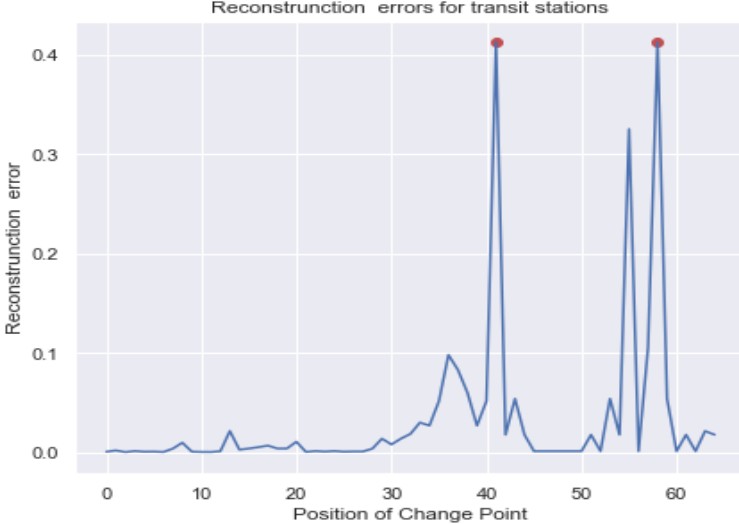

**Figure 7.** Reconstruction errors for transit stations.

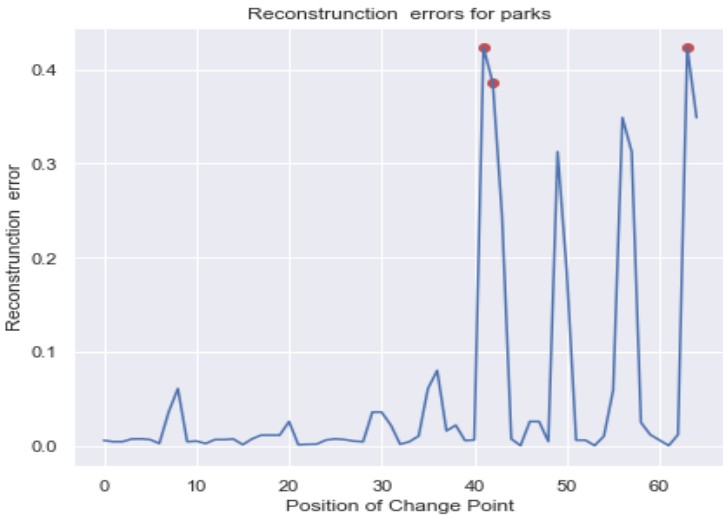

**Figure 8.** Reconstruction errors for parks.

The horizontal axis shows the positions/locations of the change-points detected by the hybrid LSTMAE model. Thereafter, the detected change-points are matched with the exact dates of the interventions. 15 February 2020, corresponds to position 1 and 19 April 2020, corresponds to position 65. The change-points are shown as dots at the points where the reconstruction errors are highest. Figures 4 and 6–9 all contain the change-point detected at location 42 (27 March 2020). For grocery and pharmacy, the change-point was detected at location 41, i.e., a day before the start of the lockdown. The change-points at locations 41 and 42 coincided with the first major intervention, a full lockdown (lockdown Level 5) that was imposed by the government of South Africa on 27 March 2020. This shows that our hybrid model accurately detected a known intervention (change-point) on 27 March 2020.

A comparison of the change-points that were detected by LSTMAE and KQE and the commonly used **R** packages namely: change-point, Bayesian change-point (bcp), strucchange (breakpoints), from 15 February 2020 to 19 April 2020 is shown in Table 1.

**Table 1.** A comparison of the date of occurrence and location of the change-points that were detected by the different algorithms between 15 February 2020 and 19 April 2020, inclusive.

| Category of Place | Method | | | |
|---|---|---|---|---|
| | LSTMAE + KQE | Changepoint | bcp | Strucchange |
| Grocery and pharmacy | 26/03/2020 (41), 10/04/2020 (56) | 26/03/2020 (41) | 23/03/2020 (38), 25/03/2020 (40), 26/03/2020 (41) | 26/02/2020 (12), 26/03/2020 (41) |
| Parks | 27/04/2020 (42), 18/04/2020 (64) | 26/03/2020 (41) | 26/03/2020 (41) | 26/02/2020 (12), 26/03/2020 (41) |
| Retail and recreation | 27/04/2020(42), 10/04/2020 (56), 13/04/2020 (59) | 26/03/2020 (41) | 24/03/2020 (39), 26/03/2020(41) | 24/02/2020 (11), 07/03/2020 (22), 26/03/2020 (41) |
| Residential | 27/04/2020(42), 10/04/2020 (56), 13/04/2020 (59) | 26/03/2020 (41) | 26/03/2020 (41) | 15/03/2020 (30), 26/03/2020 (41) |
| Workplaces | 27/04/2020(42), 10/04/2020 (56), 13/04/2020 (59) | 26/03/2020 (41) | 26/03/2020 (41) | 15/03/2020 (30), 26/03/2020 (41) |
| Transit stations | 27/04/2020(42), 10/04/2020 (56), 13/04/2020 (59) | 26/03/2020 (41) | 09/03/2020 (25), 20/03/2020 (35), 26/03/2020 (41) | 25/02/2020 (11), 14/03/2020 (29), 26/03/2020 (41) |

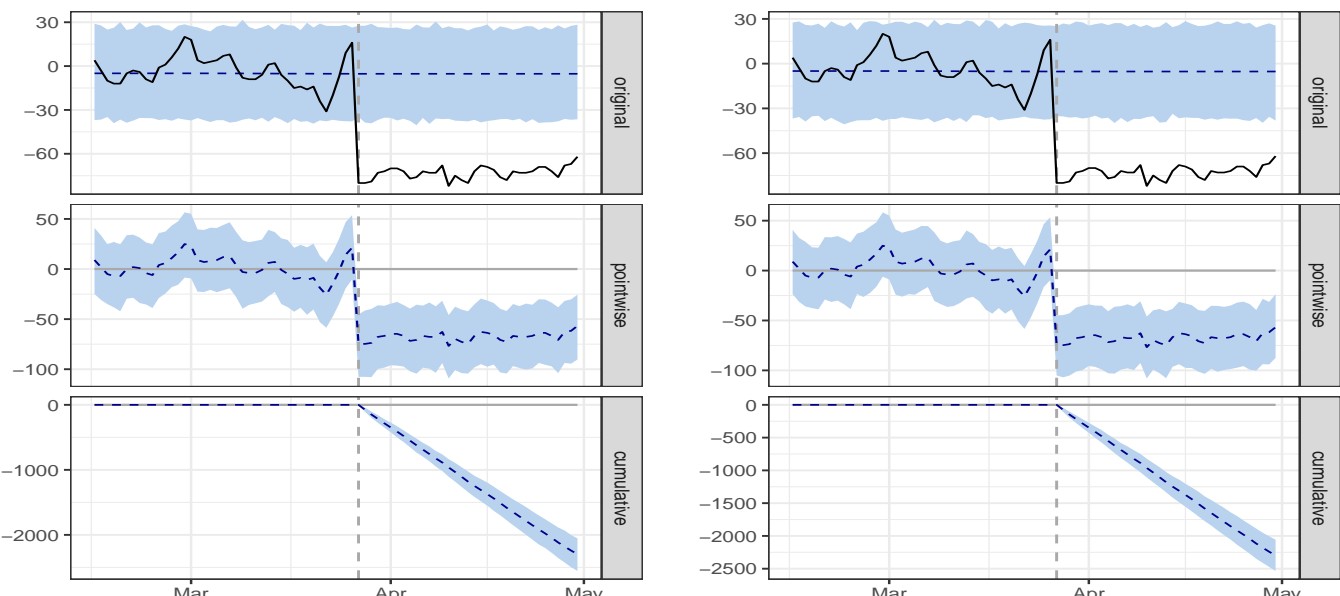

**Figure 9.** **Left**: Effect of the national full lockdown Level 5 on grocery and pharmacy **Right**: Effect of the national full lockdown Level 5 on retail and recreation.

We note that the change-point at location 42 is detected as an "anomaly" by the LSTMAE and KQE algorithm. This means that our proposed algorithm was able to detect a change in population mobility at location 42. This change-point corresponds to the first major full lockdown that was imposed on 27 March 2020, for an initial period of 21 days to curb the spread of COVID-19. For grocery and pharmacy, our proposed algorithm detected a change in mobility at location 41 on 26 March 2020. The increased population mobility on this date was a result of people stocking up on grocery and pharmacy supplies in preparation for the full lockdown. The change-point, Bayesian change-point (bcp), strucchange (breakpoints) methods detected a change-point at location 41 on 26 March 2020 as shown in Table 1. These methods measure changes in the statistical properties before and after a change-point. This means that the statistical properties of the observations from location 1 (15 February 2020) to location 41 (26 March 2020) and the statistical properties of the observations from location 43 (28 March 2020) to location 65 (19 April 2020) are different in the mean, therefore making location 42 on 27 March 2020, a change-point.

Besides, our proposed model, the LSTMAE and KQE algorithm, detected more change-points after 27 March 2020. For example, a change-point was detected at location 56 (10 April 2020) for retail and recreation, grocery and pharmacy, and workplaces and location 59 (13 April 2020) for workplaces and transit stations, 64 (18 April 2020) for parks. The change points detected by the LSTMAE + KQE at locations 56 and 59 were picked up anomalies indicating unusual mobility around those dates. This increase in mobility on position 56 (10 April 2020) and 59 (13 April 2020) could be a result of the Easter holidays for 2020. This may indicate that the movement restrictions were not effectively monitored during the Easter holidays as there was an increase in population movements on Easter Friday and Easter Monday.

We observe that the Bayesian change-point (bcp) method, detected change-points at locations 38 (23 March 2020), 40 (25 March 2020), 41 (26 March 2020), for grocery and pharmacy and 39 (24 March 2020), 41 (26 March 2020) for retail and recreation as shown in Table 1. These change-points were as a result of increased population mobility as people were stocking up on food items and other essential commodities in preparation for the full lockdown on 27 March 2020. The strucchange (breakpoints) method detected change-points at location 11 (24 February 2020) for transit stations, and retail and recreation, location 12 (25 February 2020) for parks, and grocery and pharmacy. The R package, strucchange (breakpoints) also detected a change-point at location 22 (7 March 2020) for retail and

recreation. For locations 11 and 12, the change-points can be attributed to the fact that most people get paid their salaries or wages on the 25th of each month in South Africa. The increased population mobility is therefore as a result of people moving around to access their salaries. The change-point at location 22 may be as a result of people moving around to access the South African Social Security Agency (SASSA) grants that are normally paid around that time. It is interesting to note that, at locations 11, 12 and 22, the LSTMAE and KQE algorithm did not detect any anomaly or change-point in population mobility across all the category of places at locations. This means that our proposed model did not detect any anomalous behaviour in the population movements from 15 February 2020, to 26 March.

*5.2. Change-Point Detection Using Different Datasets*

To check for the robustness of the LSTMAE and KQE algorithm, change-point, Bayesian change-point (bcp), and strucchange (breakpoints), in detecting possible change points beyond 19 April 2020, we applied these methods to two additional non-overlapping datasets from 20 April 2020, to 18 May 2020. as well as from 19 May 2020, to 19 June 2020. The locations of the change-points that were detected by these methods from 20 April 2020, to 18 May 2020, are shown in Table 2. The results show that the LSTMAE and KQE detected a change point at location 73 on 26 April 2020 for residential, workplaces and transit stations by the LSTMAE and KQE algorithm. Most people get paid their salaries or wages on the 25th of each month in South Africa. The increased population mobility is therefore a result of people moving around to access their salaries. The LSTMAE and KQE, changepoint and bcp algorithms detected a change-point at either location 76 (30 April 2020) or location 77 (1 May 2020) (Table 2).

**Table 2.** A comparison of the date of occurrence and location of the change-points that were detected by the different algorithms between 20 April 2020, and 18 May 2020, inclusive.

| Category of Place | Method | | | |
|---|---|---|---|---|
| | **LSTMAE + KQE** | **Changepoint** | **bcp** | **Strucchange** |
| Retail and recreation | 01/05/2020 (77) | 30/04/2020 (76) | 30/04/2020 (76) | 27/04/2020 (73), 03/05/2020 (79) |
| Grocery and pharmacy | 30/04/2020 (76) | 30/04/2020 (76) | 30/04/2020 (76) | 30/04/2020 (76) |
| Residential | 27/04/2020 (73), 08/05/2020 (84) | 30/04/2020 (76) | 30/04/2020 (76) | 30/04/2020 (76) |
| Workplaces | 27/04/2020 (73), 01/05/2020 (77) | 30/04/2020 (76) | 30/04/2020 (76) | 30/04/2020 (76), 08/05/2020 (84) |
| Parks | 01/05/2020 (77) | 30/04/2020 (76) | 30/04/2020 (76) | 27/04/2020 (73), 03/05/2020 (79) |
| Transit stations | 27/04/2020 (73), 01/05/2020 (77) | 30/04/2020 (76) | 30/04/2020 (76) | 30/04/2020 (76) |

The change points at locations 76 and 77 coincided with the gradual easing of the full lockdown by the government of South Africa [61]. The easing of the full lockdown was done to enable economic activities to gradually recover. Thus, from the 1 May 2020, the government of South Africa adopted a deliberate, risk-adjusted and careful approach to the easing of the lockdown restrictions. The country moved from Level 5, a full lockdown to lockdown Level 4 with fewer restrictions than those imposed under lockdown Level 5. Our proposed model and the strucchange method detected a change-point at location 84 (8 May 2020). At this point, the South African government implemented a staggered return to work plan [62]. Individuals were required to apply for permits to return to work. The working hours were also reviewed so that not all employees would leave or return to work at the same time.

Table 3 shows the locations of the change-points that were detected by the different methods between 19 May 2020, to 19 June 2020.

**Table 3.** A comparison of the date of occurrence and location of the change-points that were detected by the different algorithms between 19 May 2020, to 19 June 2020, inclusive.

| Category of Place | Method | | | |
|---|---|---|---|---|
| | LSTMAE + KQE | Changepoint | bcp | Strucchange |
| Workplaces | 31/05/2020 (107) | 28/05/2020 (104) | 24/05/2020 (100), 28/05/2020 (104) | 28/05/2020 (104), 19/06/2020 (126) |
| Parks | 30/05/2020 (106), 31/05/2020 (107) | 30/05/2020 (106) | 24/05/2020 (100), 30/05/2020 (106) | 30/05/2020 (106), 14/06/2020 (121) |
| Transit stations | 30/05/2020 (106) | 28/05/2020 (104) | 24/05/2020 (100) | 25/05/2020 (101), 27/05/2020 (103) |
| Retail and recreation | 30/05/2020 (106) | 28/05/2020 (104) | 24/05/2020 (100) | 24/05/2020 (100), 27/05/2020 (103) |
| Grocery and pharmacy | 30/05/2020 (106) | 30/05/2020 (106) | 30/05/2020 (106) | 25/05/2020 (101), 26/05/2020 (102) |
| Residential | 31/05/2020 (107), 06/06/2020 (113), 16/06/2020 (123) | 28/05/2020 (104) | 28/05/2020 (104) | 26/05/2020 (102), 28/05/2020 (104), 19/06/2020 (126) |

There are change-points that were detected at locations 100 (24 May 2020), 101 (25 May 2020), 102 (26 May 2020), 103 (27 May 2020), 104 (28 May 2020) by some of the methods as shown in Table 3. These change-points fall on the dates that most people in South Africa receive their salaries. Our proposed model detected change-points at location 106 (30 May 2020) for grocery and pharmacy, parks, transit stations, and retail and recreation, and at location 107 (31 May 2020) for workplaces, parks, and residential places. These change-points were a result of the transition from Level 4 lockdown to Level 3 lockdown on 1 June 2020. The government took a differentiated approach when dealing with hot-spot areas that had high rates of COVID-19 transmission and infections. The LSTMAE and KQE algorithm detected a change-point at location 113 (6 June 2020) for residential places which coincided with the payments of SASSA grants. The model detected a change point at location 123 (16 June 2020) for residential places. 16 June 2020, is a public holiday (Youth Day) that is recognised to commemorate the Soweto Uprising, which took place on 16 June 1976. Our proposed model was able to detect the change in population mobility on this day as people moved within residential places socialising and commemorating the day.

*5.3. Evaluating the Effect of the Full Lockdown Level 5 Effective 27 March 2020, on Population Mobility*

A full lockdown Level 5 was implemented by the government of South Africa from midnight of 26 March 2020 to 16 April 2020. There are 5 levels of the lockdown process, where Level 5 is the full lockdown imposed on 27 March 2020 and Level 1 is when the country is essentially functioning normally. Under lockdown Level 5, people were prohibited from leaving their homes, except for strict reasons (aside from essential workers in the response). People who broke the lockdown rules were either detained and/or fined as punishment for breaking the rules. We utilised the BSTSMs described in Section 3.5 to perform time series causal inference on the six categories of places. The analysis was carried out to infer the causal effect of the South African government's COVID-19 lockdowns on population mobility in these six categories of places. As noted in Section 3.1, we are restricting our analysis to changes that can be explained by the COVID-19 pandemic and the government's interventions, and not changes that reflect seasonal movements.

Figures 10–12 show the effect of the national full lockdown Level 5 that was imposed by the South African government on 27 March 2020.

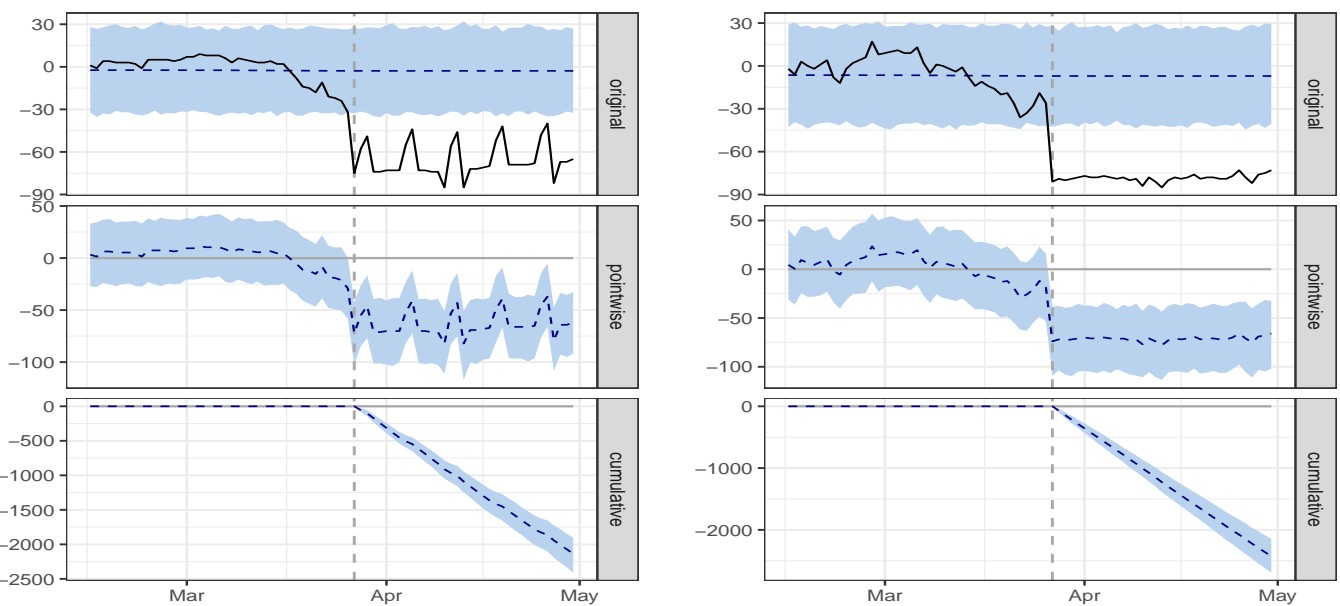

**Figure 10. Left**: Effect of the national full lockdown Level 5 on workplaces **Right**: Effect of the national full lockdown Level 5 on transit stations.

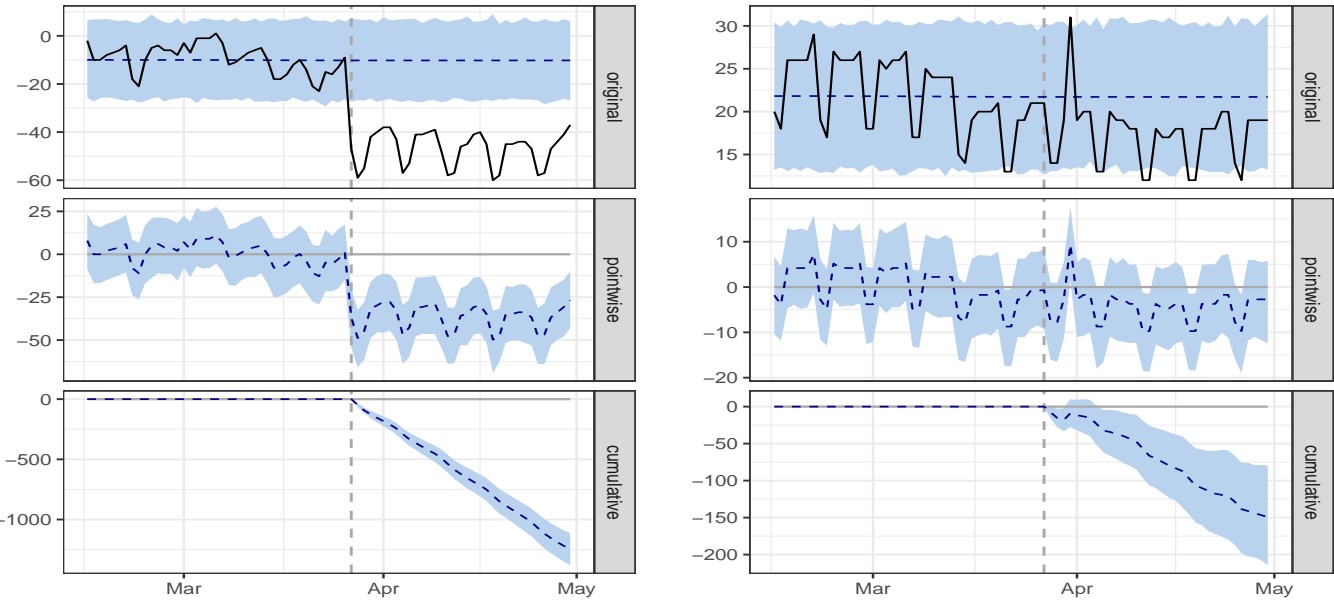

**Figure 11. Left**: Effect of the national full lockdown Level 5 on parks **Right**: Effect of the national full lockdown Level 5 on residential places.

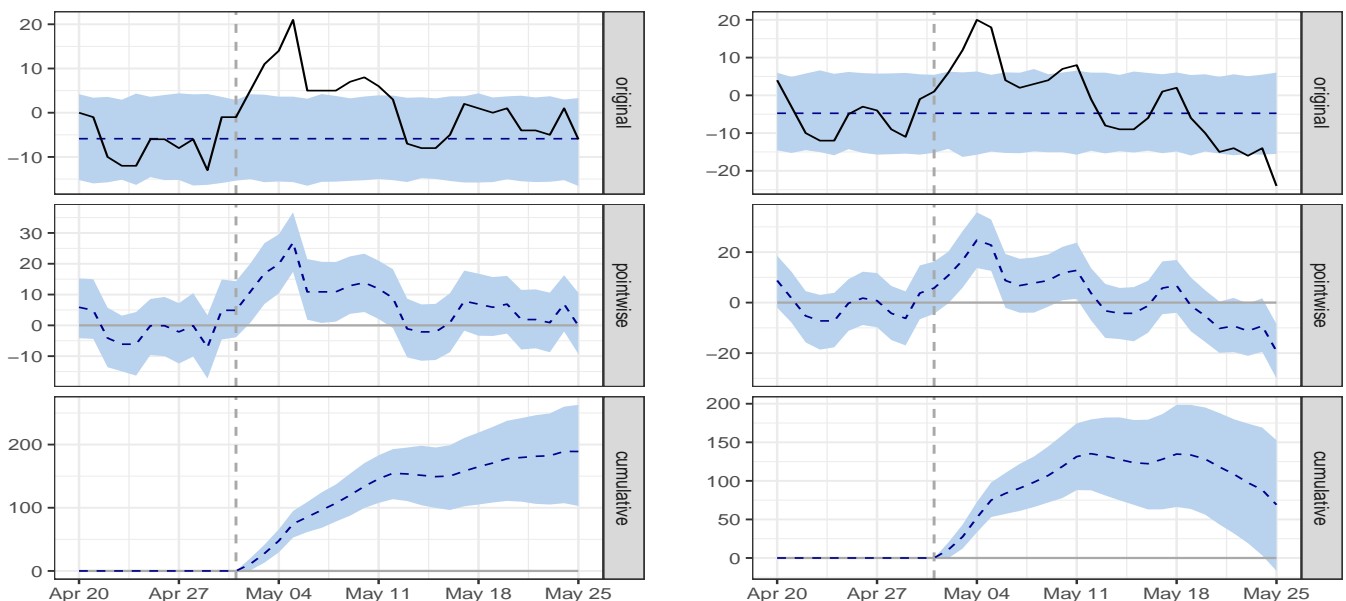

**Figure 12. Left**: Effect of the national full lockdown Level 5 on grocery and pharmacy **Right**: Effect of the national full lockdown Level 5 on retail and recreation.

The top panels in the graphs, show the counterfactual predictions represented by the dashed line and the corresponding confidence interval for the counterfactual (the shaded part). The solid line represents the actual values observed after the intervention i.e., from the 27 March 2020 when the government ordered a full lockdown to 30 April 2020. The difference between the actual population mobility and the counterfactual predictions of population mobility, which represents the estimated treatment effect of the full lockdown is shown in the middle panel. The bottom panel shows a way of visualising the effects of the interventions by using a cumulative effect plot. The plot shows the cumulative treatment effect up to that day.

A visual inspection of the graphs clearly shows a change-point in the population mobility data in all the categories of places. The full lockdown imposed by the government of South Africa on 27 March 2020, resulted in much lower movements of people in the categorised places than before the full lockdown. The estimates of the causal effect of the full lockdown imposed on 27 March 2020, for each category of places are shown in Table 4.

The actual column shows the average (across time) population mobility during the pre-intervention period (15 February 2020 to 26 March 2020). The predicted column shows the predicted counterfactual during the post-intervention period which indicates how the population movements would have behaved without the lockdown in place. For example, during the post-lockdown period, the population mobility for grocery and pharmacy was approximately equal to an average actual value of −46. However, an average predicted or counterfactual value of 0.27 would have been obtained in the absence of an intervention. The causal effect estimate column is the estimated average causal effect of the lockdown. An estimate of the causal effect the lockdown on the response variable is found by subtracting the predicted (counterfactual) average value from the actual average value. Thus, for grocery and pharmacy, the causal effect of the lockdown on population mobility is −46.27, with a 95% posterior confidence interval of [−54, −39]. These results show that there was a decrease in population mobility in places of grocery and pharmacy after the lockdown compared to the baseline days. Since the 95% posterior confidence interval does not include 0, we conclude that the lockdown imposed on 27 March 2020 had a causal effect on population mobility in grocery and pharmacy places. In relative terms, population mobility in grocery and pharmacy places decreased by −17,137.04%—from a predicted 0.27 to an actual—46. The 95% interval of this percentage is [−19,850%, −14,306%] with a Bayesian

one−sided $p$−value = 0.001 < 0.05. This means that the probability of obtaining the causal effect by chance is very small. Thus, the causal effect is statistically significant.

Similarly, Table 4 shows that in relative terms, population movements at transit stations, retail and recreation, workplaces, and parks had decreased and were all significant at 5% level of significance The results show that there are smaller changes in population mobility for residential places compared to the other categories. The data for residential places shows how the time spent at home changes. In contrast, other places show how the total number of visitors change (https://ourworldindata.org/covid-mobility-trends) (accessed on 23 November 2021).

*5.4. Evaluating the Effect of Lockdown Level 4 Effective 1 May 2020 on Population Mobility*

On 1 May 2020, the full lockdown Level 5 imposed on 27 March 2020 was gradually eased. South Africa began a measured and phased recovery of economic activity. The country implemented a risk-adjusted strategy through which the government took a thoughtful and careful approach to the easing of lockdown restrictions imposed on 27 March 2020. We assessed whether our proposed algorithm was able to detect the change or transition from full lockdown Level 5 to a Level 4 lockdown. Under lockdown Level 4, movement restrictions were eased, and all South Africans were required to wear a face mask whenever they left their homes. Some businesses could resume operations under specific conditions. However, the government encouraged businesses to implement work-from-home strategies where possible. Some activities such as walking, jogging, and cycling were permitted between 9 am and 6 am. Figures 13–15 show the effects of easing the lockdown from Level 5 to Level 4. A visual inspection of the graphs clearly shows that the easing of the full lockdown Level 5 to Level 4 on 1 May 2020, resulted in an increase in the movements of people in most places, except for residential places that showed no change in population movements.

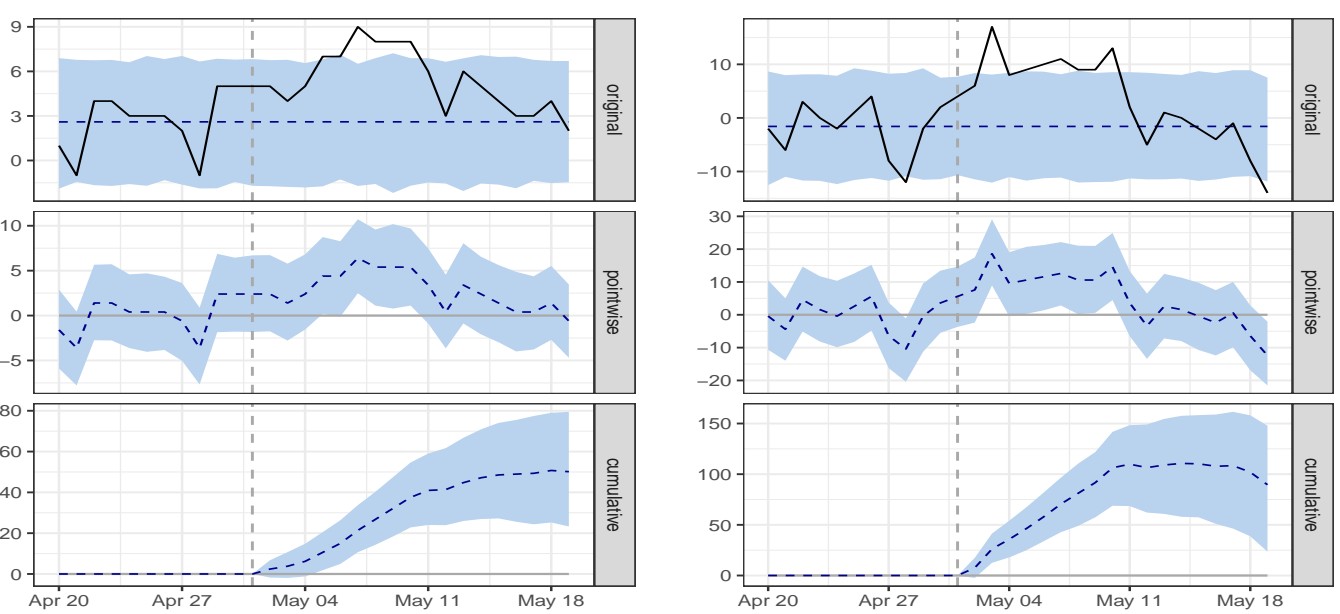

**Figure 13. Left**: Effect of the national full lockdown Level 5 on workplaces **Right**: Effect of the national full lockdown Level 5 on transit stations.

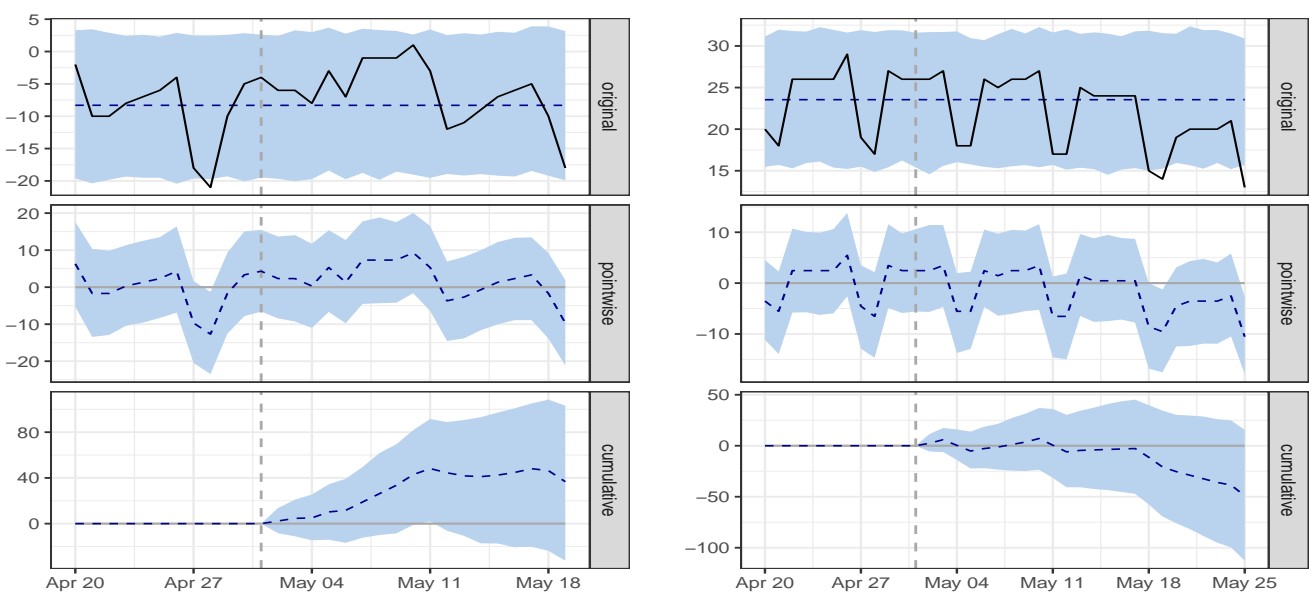

**Figure 14. Left**: Effect of the national full lockdown Level 5 on parks **Right**: Effect of the national full lockdown Level 5 on residential places.

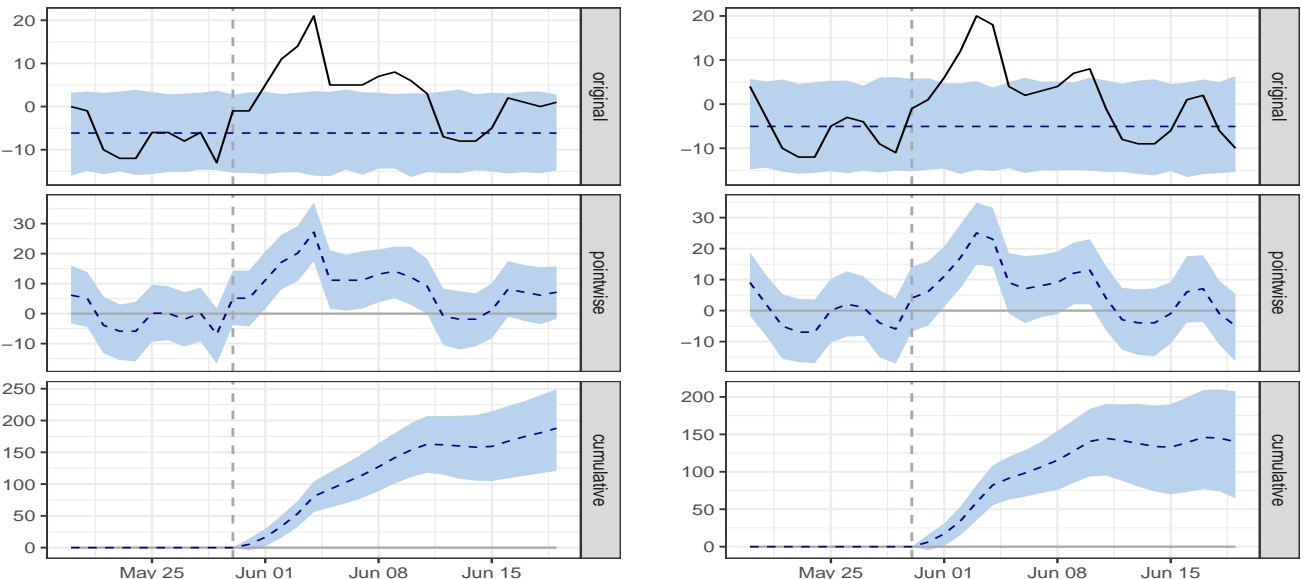

**Figure 15. Left**: Effect of the national full lockdown Level 5 on grocery and pharmacy **Right**: Effect of the national full lockdown Level 5 on retail and recreation.

The causal effect estimates of transitioning from lockdown Level 5 to lockdown Level 4 are shown in Table 5. In relative terms, population movements at retail and recreation places, transit stations, grocery and pharmacy places, and workplaces, increased. The causal effect estimates were all significant at 5% level of significance. Therefore, we conclude that the changeover from full lockdown Level 5 to Level 4 on 1 May 2020, influenced population mobility in these categories of places. The results show that, for residential places and parks, the population movements under lockdown Level 4 did not significantly change from the population movements under lockdown Level 5. For example, under lockdown Level 4, public parks, nature reserves, and beaches remained closed, hence the insignificant change in population mobility.

*5.5. Evaluating the Effect of Lockdown Level 3 Effective 1 June 2020 on Population Mobility*

On 1 June 2020, South Africa was moved from lockdown Level 4 to lockdown Level 3. The government took a differentiated approach to deal with COVID-19 hotspot areas that had far higher levels of infection and transmission. Some of the measures taken by the government, included allowing wholesale and retail trades (including stores, spaza shops and informal traders) to fully open. Additionally, universities could safely accommodate no more than a third of the student population on campus. Under lockdown Level 3, all manufacturing, mining, construction, financial services, professional and business services, information technology, communications, government services and media services could operate subject to hygiene and social distancing measures. Figures 16 and 17 show the effect of the changeover from lockdown Level 4 to lockdown Level 3. A visual inspection of the graphs clearly show that the easing of the full lockdown Level 4 to Level 3 on 1 June 2020 resulted in an increase in the movements of people in most places, except for residential places that showed no change in population movements.

The causal effect estimates of the changeover from lockdown Level 4 to lockdown Level 3 is shown in Table 6. In relative terms, there was a significant increase in the number of visitors to places like grocery and pharmacy, retail and recreation, workplaces, and transit stations. Therefore, we conclude that the changeover from lockdown Level 4 to Level 3 on 1 June 2020, influenced population mobility in these categories of places. However, for residential places and parks, the population movements did not significantly change from the mobility trends witnessed under lockdown Level 4.

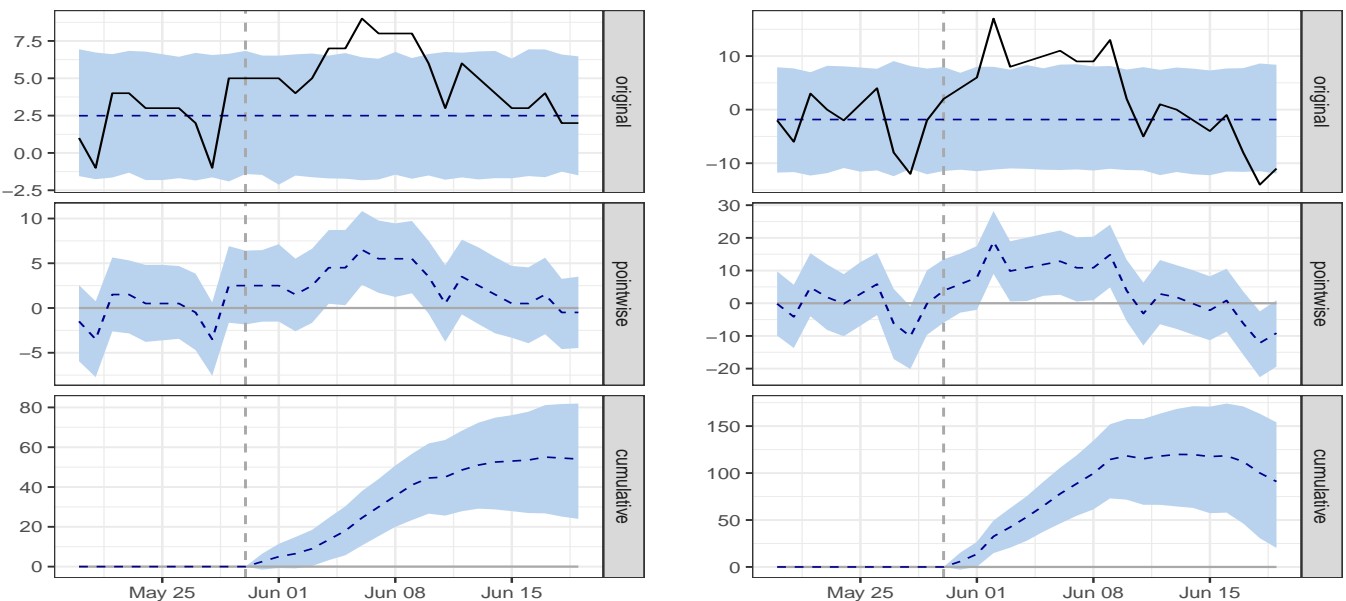

**Figure 16. Left**: Effect of the national full lockdown Level 5 on workplaces **Right**: Effect of the national full lockdown Level 5 on transit stations.

**Table 4.** Causal effect of lockdown Level 5 implemented 27 March 2020 for each category of places.

| Category of Place | Actual | Predicted | Causal Effect Estimate | 95% CI | Relative Effect | 95% CI | Bayesian One-Sided $p$−Values |
|---|---|---|---|---|---|---|---|
| grocery and pharmacy | −46 | 0.27 | −46.27 | [−54, −39] | −17,137.04% | [−19,850%, −14,306%] | 0.001 |
| retail and recreation | −73 | −5.3 | −67.7 | [−75, −60] | −1277.36% | [−1417%, −1136%] | 0.001 |
| Workplaces | −66 | −2.9 | −63.1 | [−70, −55] | −2175.86% | [−2452%, −1928%] | 0.001 |
| Parks | −47 | −10 | −37 | [−40, −33] | −370.00% | [−395%, −319%] | 0.001 |
| Transit Stations | −78 | −7.1 | −70.9 | [−80, −63] | −998.59% | [−1130%, −893%] | 0.001 |
| Residential | 17 | 22 | −5 | [−6.4, −2.4] | −22.73% | [−30%, −11%] | 0.001 |

**Table 5.** Causal effect of lockdown Level 4 implemented 1 May 2020, for each category of places.

| Category of Place | Actual | Predicted | Causal Effect Estimate | 95% CI | Relative Effect | 95% CI | Bayesian One-Sided $p$−Values |
|---|---|---|---|---|---|---|---|
| grocery and pharmacy | 3.6 | −5.9 | 9.5 | [6.1, 13] | 161.02% | [103%, 217%] | 0.001 |
| retail and recreation | 2.7 | −4.8 | 7.5 | [3.5, 11] | 156.25% | [72%, 232%] | 0.001 |
| Workplaces | 5.4 | 2.6 | 2.8 | [1.3, 4.4] | 107.69% | [50%, 170%] | 0.001 |
| Parks | −6.3 | −8.3 | 2 | [−1.8, 5.7] | 24.10% | [−69%, 22%] | 0.158 |
| Transit Stations | 3.4 | −1.6 | 5 | [1.3, 8.2] | 312.50% | [82%, 518%] | 0.001 |
| Residential | 22 | 24 | −2 | [−4.7, 0.65] | 8.33% | [−20%, 2.8%] | 0.063 |

**Table 6.** Causal effect of lockdown Level 3 implemented 1 June 2020, for each category of places.

| Category of Place | Actual | Predicted | Causal Effect Estimate | 95% CI | Relative Effect | 95% CI | Bayesian One−Sided $p$−Values |
|---|---|---|---|---|---|---|---|
| grocery and pharmacy | 3.2 | −6.1 | 9.3 | [6.3, 13] | 152.46% | [103%, 205%] | 0.001 |
| retail and recreation | 1.9 | −5.1 | 7 | [3.2, 10] | 137.25% | [64%, 205%] | 0.001 |
| Workplaces | 5.2 | 2.5 | 2.7 | [1.2, 4.1] | 108.00% | [48%, 164%] | 0.001 |
| Parks | −6.8 | −8.4 | 1.6 | [−2.5, 5.8] | 19.05% | [−69%, 30% ] | 0.209 |
| Transit Stations | 2.7 | −1.8 | 4.5 | [1, 7.7] | 250.00% | [55%, 417%] | 0.007 |
| Residential | 22 | 23 | −1 | [−4.1, 1.6] | 4.35% | [−18%, 7%] | 0.213 |

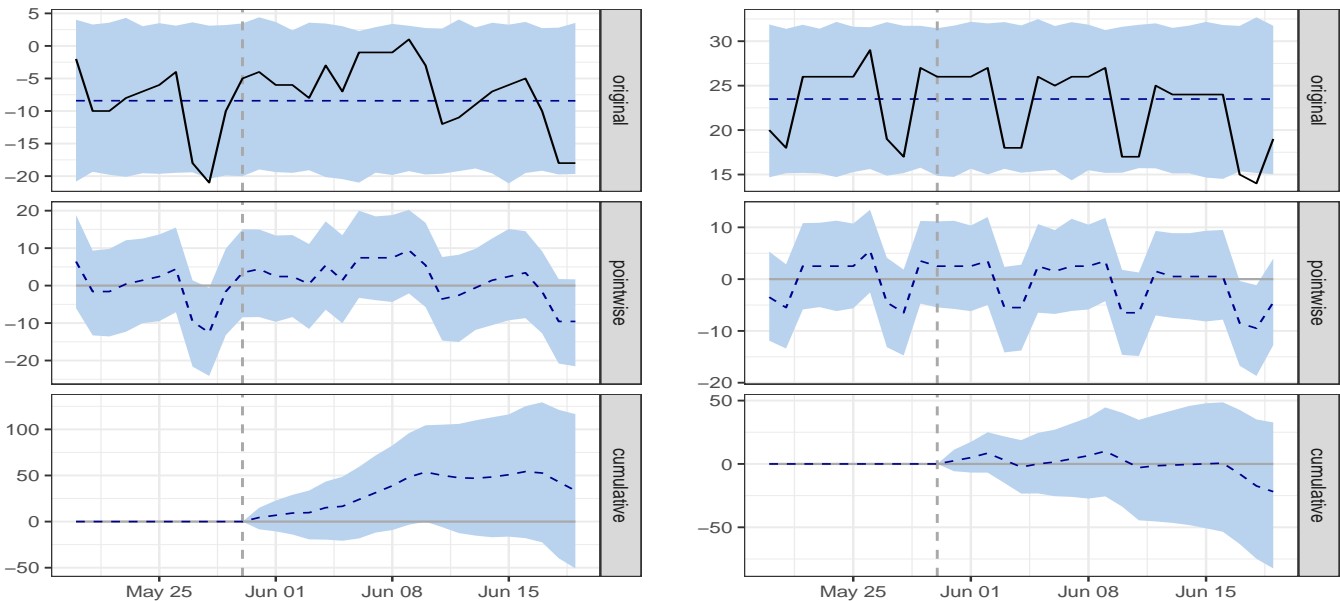

**Figure 17. Left**: Effect of the national full lockdown Level 5 on parks **Right**: Effect of the national full lockdown Level 5 on residential places.

## 6. Discussion and Conclusions

### 6.1. Discussion

In this paper we have, 1. successfully developed a model that integrates a long short-term memory network (LSTM) and a kernel quintile estimator (KQE) to detect change-points in time series or sequential data; 2. developed a non-parametric that does not require advanced knowledge of the true number of change-points; 3. developed a model that can detect abrupt changes such as lockdowns, that are sufficiently "large" regardless of the noise levels in the data and the size of the data. This is crucial to avoid having several false positives.

We have used a data point's reconstruction error, which is the error between the data point's original value and its low dimensional reconstruction, to detect a change-point as an anomaly. Change-point detection methods that detect changes in parameters such as the mean or variance do not detect isolated abnormal points such as anomalies, and they should be supplemented with a Shewhart control [8]. Our algorithm addresses this shortcoming as the change-points are detected as anomalies in the time series and the algorithm does not depend on the statistical properties such as the mean before or after a change-point. The key factor of the performance of reconstruction-based methods is the threshold, which represents the value of the reconstruction error where a data point is labeled as an anomaly or change-point. Thus, we do not estimate changes in the mean process or the changes in the mean and/or variance of a classical model. However, work has been done in the past to detect change-points in model parameters [23,63–65]. If the underlying functional form is correctly specified, then the parametric techniques become efficient. Our proposed model does not make strong assumptions about a specific functional form; thus, the model can freely learn any functional form from the training data.

Our method, the LSMAE and KQE was successful in determining the number and exact time of the major change-points in the population mobility during the period from 15 February 2020, to 31 March 2020. The proposed model successfully detected the change-point as a result of the full lockdown Level 5 that was imposed by the South African on 27 March 2020. We used other datasets beyond 30 April 2020, to determine if our model was able to capture other different levels of the lockdown. Using a dataset from 20 April 2020,

to 25 May 2020, our model successfully detected the changeover from lockdown Level 5 to lockdown Level 4 on 1 May 2020. We used another dataset from 20 May 2020, to 19 June 2020, and our proposed model successfully detected the changeover from lockdown Level 4 to Level 3. This means that our model was successful in capturing some of the interventions (in this case lockdowns) that were imposed by the government of South Africa from 15 February 2020 to 19 June 2020.

In this paper, an approach to inferring the causal effect of COVID-19 interventions has been proposed. The approach uses a hybrid model that incorporates an LSTM autoencoder and a kernel quantile estimator to detect change-points that are then used to infer the causal effects of the COVID-19 interventions. We implemented our model to detect change points using time series data on population mobility trends before and after an intervention. We used the BSTMs that are implemented in the CausalImpact R package to predict the counterfactual. The causal effect was estimated as the difference between the observed population mobility (before the intervention) and the population mobility that would have been observed had the intervention not taken place (counterfactual).

The lockdown imposed by the government of South Africa on 20 March 2020, did cause a significant decrease of activities in all the categorised places as shown in Table 4. These findings about the causal effects of the lockdown adds to emerging evidence that interventions such as lockdowns significantly reduce mobility [66–68]. These findings suggest that the causal effects of interventions on population mobility need to be strongly considered before taking measures that can severely affect the people's livelihoods and the economy. For example, [67] found out that lockdowns disproportionately affect the poor in a country. Ref. [69] states that measures that are taken by countries against the spread of COVID-19 often bring along unprecedented economic hardships. The changeover from lockdown Level 5 to lockdown Levels 4 and 3, did cause a significant increase of activities in transit stations, grocery and pharmacy, retail and recreation, and workplaces, except for residential places and parks which showed no significant changes during the transitions as shown in Tables 5 and 6. Parks remained closed during lockdown Level 4 and only a limited number of open-access national parks could open under lockdown Level 3. For residential places, the insignificant change in population mobility because people equally spend more time at home even on workdays.

Making inferences about the effect of COVID-19 interventions is a crucial process that must be done in a timely manner. This is because understanding the effects of such measures can inform policymakers to make the right decisions. If policymakers think that imposing interventions results in little effect, they may be faced with a situation where infections may rise again [44]. On the other hand, if the policymakers believe that the interventions they impose may significantly slow down the spread of COVID-19, then the interventions may be maintained for longer periods. However, this damages economic and social recovery, and it is vital to strike a balance between the potential positive effects of population mobility restrictions on public health and the potential negative social and economic impacts.

### 6.2. Limitations

A possible limitation of our proposed approach is that it was not evaluated in high-dimensional settings (the so-called curse of dimensionality). Therefore, we were not able to determine its accuracy in high-dimensional settings. Another limitation is that the ACAPS COVID-19 Government Measures dataset [7] used in this paper only contained a description of the measures taken by the government of South Africa from 10 March 2020, to 7 July 2020. This means that our analysis only covered the three lockdown measures (Levels 5, 4 and 3) as they fall within the period of 10 March 2020, to 7 July 2020. The Google COVID-19 community mobility reports do not provide the actual number of people and duration of stay values as well as the median values. They only show how the number of people or duration of stay has changed relative to the median which is a limitation.

For future work we would like to evaluate our method in (1) high-dimensional settings, (2) detecting multiple change-points in multivariate time series or genomic sequences, (3) identifying possible mutations in SARS CoV-2 genomic sequences and evaluate the causal effect of the identified mutations and (4) finding the relationship between population mobility and the rate of transmission of the virus.

## 7. Conclusions

We have used a data-driven counterfactual approach to evaluate interventions that guide governments in controlling the spread of COVID-19. The paper has made two very important contributions. Firstly, we used a deep learning approach coupled with a kernel quantile estimator to successfully detect change-points in time series data. Secondly, we performed a careful causal analysis to learn about the effects of different government interventions. The findings show that the full lockdown that was imposed on 27 March 2020, to contain the spread of COVID-19 affected population mobility and significantly reduced economic activities in transit stations, grocery and pharmacy, retail and recreation, workplaces, and parks. The findings show that people generally stayed at home during the lockdown. Currently, there is no study available in South Africa that incorporates change-point analysis on population mobility trends and causal inference to quantify the effects of an intervention such as a full lockdown on population movements.

**Author Contributions:** Conceptualization, A.W. and C.C.; methodology, A.W.; software, A.W.; validation, A.W., formal analysis, A.W.; investigation, A.W.; resources, A.W.; data curation, A.W.; writing—original draft preparation, A.W.; writing—review and editing, A.W.; visualization, A.W.; supervision, C.C. Both authors have read and agreed to the published version of the manuscript.

**Funding:** This research received no external funding.

**Data Availability Statement:** Publicly available datasets were analyzed in this study. This data can be found here: COVID-19 GOVERNMENT MEASURES DATASET: https://www.acaps.org/covid-19-government-measures-dataset (accessed on 22 November 2020) and Google COVID-19 Community Mobility Reports: https://www.google.com/covid19/mobility/ (accessed on 22 November 2020).

**Conflicts of Interest:** The authors declare no conflict of interest.

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
