# Peer review of "A Machine Learning Evaluation of the Effects of South Africa’s COVID-19 Lockdown Measures on Population Mobility"

_make, doi:10.3390/make3020025_

Round 1

Reviewer 1 Report

This paper provides estimation of change points around COVID-19 lockdown periods in South Africa by proposing a new deep learning based method (LSTMAE + KQE), and compares its performance with three alternative approaches that are based on maximum likelihood estimation, Bayesian analysis and detecting structural changes in linear models. The authors further infers the casual effects of lockdown on population mobility in different settings based on Bayesian structural time-series models (BSTSM). This paper has the potential to be a very nice one after extensive revision.

Major:

  1. The authors developed the LSTMAE + KQE method that estimated the change points better than the alternative methods including BSTSM, but used BSTSM for casual inference. Can they use LSTMAE + KQE method to perform casual inference, such as by considering “difference-in-difference” strategy?
  2. Table 1: the alternative methods detected a few change points that were not captured by the proposed method, and the authors claimed that those were “normal” activities to justify the superior performance of the proposed method than alternative methods. The authors should set up better simulation settings to demonstrate the control over false positives/negatives. For example, the authors could shuffle the days prior to lock down so no change points exist, and apply different methods to this data to compare if they detect false positives.
  3. Similarly in Table 1, the proposed method detected a few points post lockdown that were missed by alternative methods, which were also used to justify the proposed method was a better one as “extension” of policies were implemented on those days. However, shouldn’t the “extension” of an original policy keep the trend, instead of creating a change point?
  4. The paper is wordy. Can the authors tighten the manuscript, and move some non-essential pieces to supplementary materials? For example, the details of maximum likelihood estimation and structural changes in linear models can go there.

Minor:

  1. Table 1-3: the change points are shown, but it’s hard to interpret these points without context. Can the authors change them to the actual dates?
  2. Table 4-6: How is mobility score defined? What’s the unit? How can one interpret the mobility score? For example, when actual mobility is 10, is this a big or small number that are important for policy makers?

Reviewer 2 Report

I suggest including the results of the experiment in the abstract.

I suggest that the images of the results be modified as they are not of good quality.
